# Catastrophic health care spending in managing type 2 diabetes before and during the COVID-19 pandemic in Tanzania

**Peter Binyaruka** [ID]**\*, Sally Mtenga**

Department of Health System, Impact Evaluation, and Policy, Ifakara Health Institute, Dar es Salaam, Tanzania

\* pbinyaruka@ihi.or.tz

## Abstract

COVID-19 disrupted health care provision and access and reduced household income. Households with chronically ill patients are more vulnerable to these effects as they access routine health care. Yet, a few studies have analysed the effect of COVID-19 on household income, health care access costs, and financial catastrophe due to health care among patients with type 2 diabetes (T2D), especially in developing countries. This study fills that knowledge gap. We used data from a cross-sectional survey of 500 people with T2D, who were adults diagnosed with T2D before COVID-19 in Tanzania (March 2020). Data were collected in February 2022, reflecting the experience before and during COVID-19. During COVID-19, household income decreased on average by 16.6%, while health care costs decreased by 0.8% and transport costs increased by 10.6%. The overall financing burden for health care and transport relative to household income increased by 32.1% and 45%, respectively. The incidences of catastrophic spending above 10% of household income increased by 10% (due to health care costs) and by 55% (due to transport costs). The incidences of catastrophic spending due to health care costs were higher than transport costs, but the relative increase was higher for transport than health care costs (10% vs. 55% change from pre-COVID-19). The likelihood of incurring catastrophic health spending was lower among better educated patients, with health insurance, and from better-off households. COVID-19 was associated with reduced household income, increased transport costs, increased financing burden and financial catastrophe among patients with T2D in Tanzania. Policymakers need to ensure financial risk protection by expanding health insurance coverage and removing user fees, particularly for people with chronic illnesses. Efforts are also needed to reduce transport costs by investing more in primary health facilities to offer quality services closer to the population and engaging multiple sectors, including infrastructure and transportation.

## Introduction

COVID-19 is a potentially deadly disease affecting health, social welfare and economic development. The economic effects are exacerbated by a reduced workforce, job loss, and

**Data Availability Statement:** Data is available on request. There are ethical restrictions on sharing the data set publicly, one reason being data containing sensitive patient information particularly

the experience with COVID-19. The restriction was imposed by the Ifakara Health Institute (IHI) Institutional Review Board (IRB). In case of data request, please contact the IRB secretary, Dr. Mwifadhi Mrisho (+255655766675, mmrisho@ihi. or.tz).

**Funding:** SM received a grant from Medical Research Council (MRC) and the National Institute for Health Research (NIHR), grant number: MR/ V035924/1. The funder played no role in the study.

**Competing interests:** The authors have declared that no competing interests exist.

inadequate productivity in the world economy [1–3]. In the health sector, the COVID-19 pandemic has caused disruptions in healthcare provision and limited healthcare access [4–8]. As a result, it increased healthcare costs, especially for patients with chronic diseases [9]. Households with chronically ill patients also face financial catastrophe and impoverishment as they manage their illnesses [10–16]. In South Africa, for instance, a quarter of 396 patients with diabetes especially the poorest faced catastrophic health spending [13]. This is because the management of chronic diseases requires routine care, monitoring, and routine drug intake, but also health services for chronically ill patients are rarely covered by health insurance and are obtained mostly from private providers.

The household welfare loss among households containing chronically ill people is likely to have increased in the COVID-19 era, either due to increased costs of accessing healthcare and/ or lower incomes because of reduced economic activity arising from COVID-19 mitigations (e.g., lockdown and limited movement) [14, 17–19]. However, countries have responded differently to curb the transmission of the SARS-CoV-2–19 virus. In East Africa, for instance, Tanzania chose not to lockdown and delayed accepting the vaccine [20–22], while Uganda and Kenya accepted the vaccine and, respectively, imposed lockdown and curfews [23, 24]. All three countries emphasized the use of masks, reduced movements, avoiding crowded places, and hand washing. Thus, we hypothesize that COVID-19 affects household welfare in three pathways—reduced household income, increased cost to access healthcare or both. We tested this hypothesis using data from Tanzania. To the best of our knowledge, this is the first study to assess the incidence of catastrophic spending on health care and transport for those with diabetes before and during the COVID-19 pandemic.

We focussed on people with type 2 diabetes (T2D), a non-communicable disease, as they are likely to be affected especially severely with the COVID-19 pandemic [10–13]. People with T2D are more vulnerable to infection by COVID-19 [25–28] due to reduced immunity [29, 30] and to become fore severely ill. They need regular care to monitor their condition, and some either stopped accessing that care or faced increased costs in doing so.

In this study, we measured the change in household income, health care access costs, and incidence of catastrophic health spending among patients with T2D in Tanzania. We further identified the characteristics that made some patients and households more vulnerable to catastrophic health spending. We then discussed the implications of our findings for costs of accessing service delivery, for example, monitoring of chronic conditions, while ensuring financial risk protection during a crisis.

## Study setting

This study was conducted in two regions (Dar es Salaam and Morogoro) out of 31 in Tanzania. Tanzania is a lower-middle income country in East Africa with a population of 61.7 million people [31]. The Tanzanian health system is financially constrained, with limited resources. In 2020/21, the government spent 11% of its total budget on health, below the Abuja declaration target of 15%; this equated to 1.8% of GDP, also below the recommended threshold of 5% for achieving universal health coverage (UHC). The per capita total spending on health was 40.3 USD in 2019/20, close to the 44 USD target recommended by WHO for provision of essential health services but far from the 86 USD recommended to achieve UHC. The Tanzanian health system is funded from multiple sources, including government spending from tax revenue (22%), donor support (34%), out-of-pocket payments (32%), and health insurance contributions (12%) in 2019/20 [32]. However, the share of out-of-pocket payments is alarming and above the recommended threshold of 15% [33], risking high levels of catastrophic and impoverishing health spending. Tanzania also faces a triple burden of diseases, due to a recent

increase of non-communicable diseases (NCD) and injuries [34]. In 2012, for example, the prevalence of diabetes (including T2D) was 9.1% [34]. The NCD and injuries accounts for 41% of all disability-adjusted life years in Tanzania, and the burden has almost doubled in the past 25 years [35]. Tanzania also faces poor availability of medicines for hypertension and diabetes, with government facilities faring worse than mission or private facilities. Dispensary and health centres also compare poorly in drug availability with hospitals as do rural compared to urban facilities [36]. All these contextual challenges have cost implications in managing chronic conditions, which we might anticipate would be worsened during the COVID-19 pandemic. Tanzania opted not to lockdown, but people were instructed to reduce mobility and avoid congested areas, with implications for access to health care, business, and household welfare.

## Methods

### Data sources

A cross-sectional survey was conducted among 500 people with T2D alongside other non-communicable diseases. The sample size of 500 people was estimated using the Cochran formula, assuming 50% of patients with T2D experienced disruption of care during COVID-19 [37], selecting a power of 80%, provide a margin of error of 5% with 95% confidence intervals and a non-response rate of 30%. The sample included adults (18+ years) who were diagnosed with T2D before COVID-19 in Tanzania (March 2020). Out of this sample, only 4% confirmed to have COVID-19 prior to the survey. Eligible patients were identified from health facility patient registers at the outpatient department or diabetic clinic in Dar es Salaam and Morogoro region. We sampled patients from facility registers in three hospitals and one health centre per region. All patients who were eligible were contacted and asked to give written informed consent. Trained fieldworkers administered a structured questionnaire capturing information about the patients' characteristics, experience of seeking healthcare, monthly cost of accessing care, and household income. These data were captured in respect of two time periods, before (asked retrospectively) and during COVID-19. The tools were piloted and refined before the actual data collection in February 2022.

### Measurement of household living standard

We used the reported household total monthly income as a measure of living standards to assess financial burden and risk of financial catastrophe. This was considered in the absence of consumption and expenditure data, while acknowledging the concern of accuracy of income data in developing countries [38]. Household income was measured at two points, before and during COVID-19. Since some patients (n = 50) did not report their monthly household income, a key information in measuring financing burden, we excluded those individuals in the analysis and kept only 450 patients for the analysis. By using the sample of 450 patients/ households we generated two equal sized income subgroups as predictors of incurring financial catastrophe. We used a principal component analysis based on 39 items of household characteristics and asset ownership to generate a wealth score for each household [39, 40]. All 450 households were then ranked according to the wealth index/ score and categorised into two equal sized subgroups (the better-off and worse-off). As a robustness check, we also used a ladder scale of 1–10 to measure household economic position, whereby patients were asked to rank their households in terms of economic position in a ladder (i.e., 10 means highest economic position and 1 for lowest economic position).

### Health care and transport costs

We used two levels to measure the cost of accessing health care, transport and health care costs. These costs were also before and during COVID-19. The survey tool was used to capture these cost dimensions. Irrespective of being insured or exempted from fees (older patients), patients with T2D were asked to report their average total monthly costs for transport and health care. We thus captured a wide range of healthcare costs in managing T2D which included payments for consultation, medications, laboratory tests, inpatient costs, informal payments/ out-of-pocket payments, but excluding transport costs. Transport costs included total travel costs for two visits within a typical month. All cost data were reported in a local currency, Tanzanian shilling (TZS), (1 USD = 2300 TZS as an average exchange rate in 2022). The cost data did not account for inflation because we had a window of two years during COVID-19 that patients were referring to. However, the average annual inflation in Tanzania declined from 3.5% (2019 before COVID-19) to 3.3% (2020 during COVID-19), before bouncing back to 3.7% (2021) and 4.4% (2022) [41].

### Measuring financing burden and catastrophic expenditure

We measured financing burden relative to household income. The incidence of catastrophic health expenditure was measured as a headcount or proportion of households that spent at least 10% of total household income on health care and/or transport costs when managing T2D over time [38, 42, 43]. We estimated the incidence of catastrophic due to health care cost, transport cost, and total cost separately. The health care cost excluded the costs of health insurance premiums. The 25% and 40% thresholds on health spending relative to non-food household expenditure (capacity to pay) were not used because expenditure or consumption data were not captured.

### Factors associated with catastrophic health expenditure

A multivariate logistic model was used to identify significant patients' and household level characteristics associated with catastrophic spending, defined as spending at least 10% of their income (10%-threshold) on healthcare, transport or total. We used a binary dependent variable which took a value of one for patients who incurred catastrophic spending either before or during COVID-19 and zero otherwise. The selection of potential determinants was informed by previous literature [44–47]. We included the following patient level factors: place of residence (Dar es Salaam vs. Morogoro), gender (male vs. female), marital status (married vs. not married), age (five categories: <40, 40–49, 50–59, 60–69, and >70 years), five education categories (no education, primary, secondary and higher education), five occupation categories (formal workers, farmers, self-employed, retired and unemployed), patients with any health insurance (insured vs. not), household socioeconomic status (two subgroups of household monthly income level–the better-off and worse-off), household size as continuous variable, and presence of comorbidity (with vs. without). All analyses were performed in STATA version 16.

### Ethical issues

Ethical approval was obtained from ethic committees in Tanzania. This include the institutional ethical approval from the Ifakara Health Institute (IHI/IRB/No: 38–2021), and the national approval from the National Institute for Medical Research (NIMR/HQ/R.8a/Vol.X/ 3806). We sought written informed consent from all respondents by providing information on the objectives of the research and the procedures applied in a clear language using an information sheet.

## Results

Most patients with T2D were in the urban region (Dar es Salaam) (60%), female (66%), married (63%), 50 years and above (80%) and had completed primary or secondary education (84%) (Table 1). The average age was 57 years. The largest number were self-employed (40%) followed by farmers (20%). The average household size was 4.9 and the majority had health insurance (60%) and comorbidities (73%).

### Economic impact of COVID-19

The average monthly household income declined significantly by 16.6% from 347,373 TZS (151 USD) (pre-COVID-19) to 289,549 TZS (126 USD) (during-COVID-19), and a similar decline of 18.4% was reported in terms of rating their economic position on the 10 point ladder scale from an average of 4.9 to 4.0 (Table 2). Since the incremental changes in both measures showed a declining trend of similar magnitude in relative terms, this confirmed the use income measure in assessing financing burden and catastrophic spending.

### Changes in health care costs

The average health care costs per month for T2D declined slightly over time by 0.8% (Table 2). However, almost a half of the 450 patients did not pay for health care before (49%) and during

**Table 1. Patient socioeconomic and demographic characteristics in Tanzania (N = 450).**

| Variable | Description | n | % |
|---|---|---|---|
| **Place of residence** | Rural region (Morogoro) | 179 | 39.8% |
| | Urban region (Dar es Salaam) | 271 | 60.2% |
| **Sex** | Male | 152 | 33.8% |
| | Female | 298 | 66.2% |
| **Marital status** | Married | 283 | 62.9% |
| | Not married | 167 | 37.1% |
| **Mean age in years [SD]** | | 450 | 56.7 [9.9] |
| **Age group** | <40 years | 31 | 6.9% |
| | 40–49 years | 59 | 13.1% |
| | 50–59 years | 153 | 34.0% |
| | 60–69 years | 183 | 40.7% |
| | ≥70 years | 24 | 5.3% |
| **Education level** | No education | 29 | 6.4% |
| | Primary education | 263 | 58.4% |
| | Secondary education | 117 | 26.0% |
| | Higher education | 41 | 9.1% |
| **Occupation status** | Formal workers | 40 | 8.9% |
| | Farmers | 91 | 20.2% |
| | Self-employed | 179 | 39.8% |
| | Retired | 65 | 14.4% |
| | Unemployed | 75 | 16.7% |
| **Health insurance** | Insured | 268 | 59.6% |
| | Not insured | 182 | 40.4% |
| **Household size [SD]** | | 450 | 4.9 [2.4] |
| **Comorbidities** | With comorbidities | 327 | 72.7% |
| | Without comorbidities | 123 | 27.3% |

**Table 2. Household mean monthly income and costs to access T2D healthcare (n = 450).**

| Variable | N | Pre-COVID-19 | During-COVID-19 | Difference | % change |
|---|---|---|---|---|---|
| Monthly household income (TZS) | 450 | 347,373 | 289,549 | −57,824*** | 16.6% |
| Household economic position (TZS) | 450 | 4.9 | 4.0 | 0.9*** | 18.4% |
| **Costs for patients with T2D** | | | | | |
| Healthcare cost (TZS) | 450 | 15,408 | 15,284 | −124 | 0.8% |
| Transport cost (TZS) | 450 | 3,465 | 3,832 | 367** | 10.6% |
| Total cost (TZS) | 450 | 18,873 | 19,116 | 243 | 1.3% |

Notes: 1 USD = 2300 TZS as an average exchange rate in 2022

*** denotes significance at 1%

** at 5%, and

* at 10% level

COVID-19 (53%) (S1 Table), with higher share of zero payments among better-off patients (before COVID-19) and among worse-off patients (during COVID-19). The average monthly transport cost to access medical care for T2D increased significantly by 10.6%, but around 10% did not pay for transport with slightly higher share of zero payments among better-off patients over time (S1 Table). The overall total cost increased by 1.3%, but generally average costs for health care were larger than transport costs (Table 2).

## Effect on financing burden and catastrophic expenditure

There was increased financing burden over time for health care and transport (Table 3). For instance, on average, 8% of household income was spent on healthcare costs, which has increased significantly to 11% during COVID-19 (2.6 percentage point change), while transport cost took 2% of household income pre-COVID-19 and increased significantly to 3% during COVID-19 (1 percentage point change) (Table 3). This shows that patients increased the share of financial resources spent for health care and transport costs relative to their total household income during COVID-19.

The incidence of catastrophic expenditure increased significantly by 2.7 percentage points due to health care cost and by 2.2 percentage points due to transport costs (Table 3). On average, 27% and 30% of households with a T2D patient spent more than 10% of their income on

**Table 3. Changes in financing burden and catastrophic health spending (n = 450).**

| | Pre-COVID-19 | During-COVID-19 | Difference | % change |
|---|---|---|---|---|
| **Relative costs to household income** | | | | |
| Healthcare cost | 8.1% | 10.7% | 2.6%*** | 32.1% |
| Transport cost | 2.0% | 2.9% | 0.9%*** | 45.0% |
| Total cost | 10.2% | 13.7% | 3.5%*** | 34.3% |
| **Incidence of catastrophic spending** | | | | |
| Healthcare cost | 26.9% | 29.6% | 2.7%* | 10.0% |
| Transport cost | 4.0% | 6.2% | 2.2%*** | 55.0% |
| Total cost | 31.8% | 35.3% | 3.5%** | 11.0% |

Notes

*** denotes significance at 1%

** at 5%, and

* at 10% level

health care costs pre- and during COVID-19, respectively; while 4% and 6% of households spent more than 10% of their income on transport costs in the two periods, respectively. The absolute figures show a higher incidence of catastrophic spending due to health care than transport costs, but the relative effect seems to be higher for transport than health care costs (10% vs. 55% change from pre-COVID-19).

## Factors associated with catastrophic spending

The logistic regression results revealed that households with patients with at least primary education had lower odds of incurring catastrophic expenditure due to health care costs, but not due to transport costs. For instance, patients with a university education were significantly less likely to incur catastrophic expenditure due to health care costs [AOR = 0.16 (95% CI: 0.03–0.91)] than patients with no education (Table 4). However, education had a weak association

**Table 4. Factors associated with catastrophic health spending (n = 450).**

| Variable | Description | Catastrophic expenditure due to healthcare cost | | Catastrophic expenditure due to transport cost | | Catastrophic expenditure due to total cost | |
|---|---|---|---|---|---|---|---|
| | | AOR | [95% CI] | AOR | [95% CI] | AOR | [95% CI] |
| **Place of residence** | Rural | 0.46* | [0.21–1.03] | 2.20 | [0.58–8.28] | 0.64 | [0.31–1.32] |
| | Urban (ref.) | | | | | | |
| **Sex** | Male | 1.85* | [0.93–3.68] | 0.18*** | [0.05–0.63] | 1.02 | [0.55–1.91] |
| | Female (ref.) | | | | | | |
| **Marital status** | Married | 1.41 | [0.76–2.63] | 2.93** | [1.10–7.81] | 1.98** | [1.09–3.59] |
| | Not married (ref.) | | | | | | |
| **Age group** | <40 years (ref.) | | | | | | |
| | 40–49 years | 1.05 | [0.28–3.92] | 0.36 | [0.06–2.19] | 0.85 | [0.23–3.16] |
| | 50–59 years | 0.43 | [0.13–1.46] | 0.26 | [0.05–1.30] | 0.36* | [0.11–1.21] |
| | 60–69 years | 0.49 | [0.14–1.76] | 0.31 | [0.05–1.75] | 0.41 | [0.12–1.46] |
| | ≥70 years | 0.49 | [0.08–2.96] | 0.27 | [0.02–4.29] | 0.35 | [0.07–1.88] |
| **Education level** | No education (ref.) | | | | | | |
| | Primary education | 0.32** | [0.11–0.94] | 0.83 | [0.19–3.57] | 0.59 | [0.21–1.67] |
| | Secondary education | 0.19** | [0.05–0.66] | 1.26 | [0.23–7.00] | 0.36* | [0.11–1.15] |
| | Higher education | 0.16** | [0.03–0.91] | 1.95 | [0.11–33.80] | 0.25* | [0.05–1.23] |
| **Occupation status** | Formal workers (ref.) | | | | | | |
| | Farmers | 2.00 | [0.43–9.32] | 1.53 | [0.23–10.35] | 1.71 | [0.49–6.03] |
| | Self-employed | 1.97 | [0.47–8.26] | 0.42 | [0.06–3.13] | 0.95 | [0.29–3.13] |
| | Retired | 1.43 | [0.27–7.64] | 0.47 | [0.03–7.67] | 0.71 | [0.17–3.03] |
| | Unemployed | 2.29 | [0.46–11.33] | 1.15 | [0.13–10.28] | 1.09 | [0.28–4.32] |
| **Health insurance** | Insured | 0.06*** | [0.03–0.11] | 0.38 | [0.11–1.27] | 0.06*** | [0.03–0.12] |
| | Not insured (ref.) | | | | | | |
| **Household income** | Worse-off group (ref.) | | | | | | |
| | Better-off group | 0.44*** | [0.24–0.78] | 0.02*** | [0.00–0.22] | 0.32*** | [0.18–0.56] |
| **Household size** | | 1.03 | [0.92–1.16] | 0.94 | [0.76–1.15] | 1.00 | [0.90–1.12] |
| **Number of comorbidities** | | 0.63 | [0.33–1.20] | 0.62 | [0.23–1.68] | 0.73 | [0.40–1.35] |
| **Constant** | | 8.51 | [1.04–69.52] | 0.86 | [0.06–11.31] | 17.20 | [2.50–118.25] |

Notes: AOR = Adjusted Odds Ratio; CI = Confidence Interval; ref. = reference group

*** denotes significance at 1%

** at 5%, and

* at 10% level

with catastrophic expenditure due to combined total cost. Households with insured T2D patients were significantly less likely to incur catastrophic expenditure due to health care costs [AOR = 0.06 (95% CI: 0.03–0.11)] and total costs [AOR = 0.06 (95% CI: 0.03–0.12)] than household with uninsured patients, but health insurance ownership was not associated with catastrophic spending due to transport costs. We also found that patients in better-off households were significantly less likely to incur catastrophic spending due to health care costs [AOR = 0.44 (95% CI: 0.24–0.78)] or transport costs [AOR = 0.02 (95% CI: 0.00–0.22)] than patients in worse-off households. The patient's gender and marital status were also associated with catastrophic spending on transport. Male patients were significantly less likely than females to incur catastrophic spending on transport [AOR = 0.18 (95% CI: 0.05–0.63)], while married patients were almost three times more likely than unmarried ones to face catastrophic spending on transport [AOR = 2.93 (95% CI: 1.10–7.81)] (Table 4).

## Discussion

This study found a reduced household income, increased transport costs, and increased financing burden and financial catastrophe associated with the onset of the COVID-19 pandemic among patients with T2D in Tanzania. This confirms the hypothesis that both reduced household income and increased cost to access health care due to COVID-19 contributes to household welfare loss. It also shows that health care expenses incurred by chronically ill patients, among other factors, mediated the harm caused by COVID-19 to household welfare and increased financial hardship. The finding of reduced household income during the COVID-19 pandemic in Tanzania is in line with other studies elsewhere [14, 18, 19, 48], and this pattern is substantially explained by disruption of income-generating activities and limited mobility due to COVID-19 restrictions. Consistent with a study in Nigeria [7], we found an increase in costs of health care access, especially on transport costs, during COVID-19 pandemic, mainly because of travel restrictions and lockdown.

We also found a slight decline in health care costs for T2D, with almost half of all patients not paying for health care during the pandemic. This is partly due to reduced health care utilisation during COVID-19, as we saw an increase in patients monitoring their condition testing at home (3–7%) and a reduced share of patients testing at health facility (87–81%) in these data. A decline in health expenditure and utilisation is consistent with an observed increased share of zero payments from 49% pre-COVID-19 to 53% during COVID-19, mostly among worse-off patients who seem to be affected the most in terms of accessing health care.

The data on catastrophic spending show that T2D patients faced relatively higher expenditure on health care than transport. However, the effect on catastrophic payments during the pandemic was larger for transport than health care cost, possibly because of increased transport cost due to limited mobility resulting from COVID-19 restrictions. In attempting to avoid crowded places, some patients were possibly hiring private cars which are more expensive than taking public transport. The incidences of catastrophic spending of 27% (pre-COVID-19) and 30% (during COVID-19) among patients with T2D are higher than 13% incidence of catastrophic spending reported among diabetic patients in public hospitals in South Africa [49], but smaller than 74.3% among diabetic patients in Ethiopia [50]. Another study reported a figure of 17.8% among diabetic populations using a 40% threshold of non-food expenditure across 35 developing countries [51]. These discrepancies in catastrophic incidences are likely due to variation in costing approaches and the structure of household economic status. Our estimate prior to COVID-19 (27%) is larger than the national estimate of less than 3%, irrespective of disease conditions across Tanzanian population [52–54]. Our estimates support the view that the presence of a chronically ill person in a household and

resulting frequent hospital visits are associated with higher incidences of catastrophic health spending [11, 15, 16, 44, 46, 55–58].

Patients with health insurance were less likely to face financial catastrophe due to health care costs, which implies that health insurance offered financial protection to diabetic patients in Tanzania. However, the effect of health insurance in a general population has been mixed, as it ranges from being protective in Ghana [59], Colombia [60], and Nigeria [61], but less protective in other settings like Kenya [44] and Vietnam [62]. A study among individuals with diabetes from 35 LMICs reported that health insurance was only protective in middle-income countries but less protective in low-income countries [51]. The effectiveness of health insurance is mixed, possibly because of differential design elements and coverage across settings. Our finding that insurance is protective supports the government's move to ensure people are covered through pre-payment mechanism for financial protection and UHC [63–65]. On the other hand, we found health insurance was not associated with reduced catastrophic transport costs, which reflects how health insurance does not cover transport costs, thereby limiting protection against catastrophic transport costs.

The finding that worse-off households that include people with T2D were more likely to incur catastrophic health spending is consistent with other studies among those with diabetes [49] and general populations [44, 45, 60, 66]. Poor households are likely to face financial catastrophe because of their low ability to pay or limited affordability of services, and typically lack health insurance [67, 68]. Moreover, patients with higher levels of education were less likely to face financial catastrophe due to health care costs, which is consistent with a study among people with diabetes in Ethiopia [50] and other studies in general populations [61, 66, 69, 70].

This study expands on the existing evidence base in three ways: first, ours is the first study to assess the incidence of catastrophic health spending among people with T2D before and during the COVID-19 pandemic. We chose T2D patients as they are likely to be impacted more by the pandemic [29, 30]. Second, we were able to test three pathways—reduced household income, increased cost to access healthcare or both–that COVID-19 affected household welfare. Third, we used comprehensive measure of health care access cost by including health care cost and transport cost, since there is little attention on transport cost when analysing catastrophic spending in health care access [44, 71].

However, our study had some limitations. First, we relied on household income to measure household living standard, despite the unreliability of these data in developing countries due to inaccuracy and recall bias [38]. We collecting income data instead of expenditure and/or consumption data because of limited resources and time. However, we also employed an alternative measure of household economic position using a ladder scale, which resulted into almost similar pattern to income estimates. Second, since we did not collect expenditure or consumption data, we were unable to use 25% and 40%-thresholds for non-food expenditure (a measure of capacity to pay) [38, 42, 43]. Third, we were unable to estimate the financing burden and risk of catastrophic spending attributable to COVID-19 management or hospitalisation among T2D patients because of limited sample size (only 4% in our sample had COVID-19). Fourth, the study was carried out in only two regions in Tanzania, and so the findings may not be generalizable to all regions as well as to other countries with varied health system set-up. Fifth, despite the cross-sectional survey capturing two time periods, this still limits the scope for causal inferences.

The findings in this study have important policy implications. Our findings show that financial protection from health care cost and transport cost for people with T2D is inadequate in Tanzania. This highlights the need for policy makers to ask how to ensure financial protection in managing chronic illnesses such as T2D, especially in the context of pandemic and emergency situation. For instance, there is a need to strengthen and expand existing health

insurance and/or enforce user fee exemption, especially for people with chronic illness as an important steps towards UHC [65]. Similarly, the benefit packages in insurance schemes should be expanded to cover routine services for chronically ill people, and explore ways to accommodate transport cost which contributes significantly into financial catastrophe [44]. It is also high time to scrutinize the design of universal health insurance in Tanzania, by ensuring key elements for strengthening financial protection of all are being considered. The findings that transport costs are substantial, leading to catastrophic spending reinforces the need for a greater investment in PHC facilities in order to bring quality health services closer to population as one of the recommended routes toward UHC [72]. The investment in PHC can be through direct facility financing, facility's construction and renovation and an increase in supply of health care workers and medical commodities [71]. Strengthening PHC facilities helps to meet patients' needs and expectations and potentially reduces the time and travel cost incurred by patients bypassing closer PHC [73, 74]. Policy makers attempting to reduce transport costs should aim for collaboration in a multisectoral approach including infrastructure and transportation sectors.

## Supporting information

**S1 Table. Distribution of zero payments.**
(DOCX)

## Acknowledgments

We would like to thank all healthcare providers, health managers and all patients with type 2 diabetes who participated or facilitated to ensure a successfully fieldwork for data collection. We also thank the whole GECO research team and Prof Martin McKee for proof-reading the final manuscript.

## Author Contributions

**Conceptualization:** Peter Binyaruka.

**Formal analysis:** Peter Binyaruka.

**Funding acquisition:** Sally Mtenga.

**Methodology:** Peter Binyaruka.

**Writing – original draft:** Peter Binyaruka.

**Writing – review & editing:** Sally Mtenga.

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
