## [Decision Letter · Decision Letter 0]

6 Jun 2023

PGPH-D-23-00508

Catastrophic health care spending in managing type 2 diabetes before and during COVID-19 pandemic in Tanzania

Dear Dr. Binyaruka,

Thank you for submitting your manuscript to PLOS Global Public Health. After careful consideration, we feel that it has merit but does not fully meet PLOS Global Public Health’s publication criteria as it currently stands. Therefore, we invite you to submit a revised version of the manuscript that addresses the points raised during the review process.

Please address these minor comments by the reviewers

We look forward to receiving your revised manuscript.

Kind regards,

Ejemai Eboreime, MD, MSc, PhD

Academic Editor

Journal Requirements:

Additional Editor Comments (if provided):

Reviewers' comments:

Reviewer's Responses to Questions

**Comments to the Author**

1. Does this manuscript meet PLOS Global Public Health’s publication criteria? Is the manuscript technically sound, and do the data support the conclusions? The manuscript must describe methodologically and ethically rigorous research with conclusions that are appropriately drawn based on the data presented.

Reviewer #1: Yes

Reviewer #2: Yes

2. Has the statistical analysis been performed appropriately and rigorously?

Reviewer #1: Yes

Reviewer #2: Yes

3. Have the authors made all data underlying the findings in their manuscript fully available (please refer to the Data Availability Statement at the start of the manuscript PDF file)?

Reviewer #1: Yes

Reviewer #2: Yes

4. Is the manuscript presented in an intelligible fashion and written in standard English?

Reviewer #1: Yes

Reviewer #2: Yes

5. Review Comments to the Author

Reviewer #1: A timely manuscript presenting key findings based on primary data among a critical subgroup of the population. The study design (pre-post retrospective survey) implies a certain risk of recall bias, thus it is understandable that household expenditure was not collected and instead income (which tends to be less varying) was assessed.

Some pleasant observations in the manuscript include i) that the analysis appears to be consistent with the study objectives, and is easy to follow along. There's evidence of rigorous behind the analysis, for instance there's demonstrated understanding of the causes of changes in healthcare costs (e.g increase in proportion of patients conducting home based testing / monitoring of blood sugars post COVID) and ii) that the results are presented in easy to interpret format, showing both absolute and relative differences for instance in Table 3.

I think it may be useful, if data are available, to understand the socioeconomic distribution of those that reported no payments for health care. One limitation with the calculation of risk of catastrophic payments is that it considers all participants surveyed as a population at risk, knowing that there are those who report zero values in the numerator because they did not access health services and therefore had no opportunity to incur costs. A disaggregation of the income of those who report zero numerator values (zero healthcare costs) may support a hypothesis of increased financial barriers to access (especially post COVID).

Finally, please mention if the changes in costs pre and post COVID are adjusted for inflation of local currency over the period, or otherwise quote the average inflation between the two time points.

Thanks again to the authors for a very relevant piece of work!

Reviewer #2: REVIEW REPORT ON: Catastrophic health care spending in managing type 2 diabetes before and during COVID-19 Pandemic in Tanzania

Manuscript Number: PGPH-D-23-00508

GENERAL COMMENTS

The authors of this manuscript provided a information on catastrophic health care spending among patients with Type2 Diabetes in the pre and post COVID-19 era in Tanzania. The study highlighted the potentials of the various factors, and their contributions towards incurring catastrophic health expenditure among the Type2 Diabetes patients in their country.

Their findings contribute to further understanding of the complexity of health care financing in their country as it affects Type 2 diabetes patients and therefore relevant.

TITLE

This adequately captures the study

ABSTRACT

This adequately described their study and concise.

INTRODUCTION

This is generally adequate and provided the background for the study.

METHODOLOGY

This is generally adequate and well presented. However, the authors could improve the flow of this aspect by explaining how they classified the participants into the two income groups namely: poor and non-poor. Which instrument did they use? Providing this information is very important for reproducibility.

DISCUSSION

Generally, well presented and discussed.

However, the authors merely duplicated their results in the first paragraph of this section without comparing them to published literature on the subject. The authors would do well to address this concern. They need to relate these results to published literature or they take off this paragraph as it sounds like a duplication if no comparison is made to published literature.

REFERENCES

These are adequate.

6. PLOS authors have the option to publish the peer review history of their article (what does this mean?). If published, this will include your full peer review and any attached files.

**Do you want your identity to be public for this peer review?** For information about this choice, including consent withdrawal, please see our Privacy Policy.

Reviewer #1: No

Reviewer #2: No

---

## [Editor Report · Decision Letter 1]

13 Jul 2023

PGPH-D-23-00508R1

Catastrophic health care spending in managing type 2 diabetes before and during COVID-19 pandemic in Tanzania

Dear Dr. Binyaruka,

Thank you for submitting your manuscript to PLOS Global Public Health. After careful consideration, we feel that it has merit but does not fully meet PLOS Global Public Health’s publication criteria as it currently stands. Therefore, we invite you to submit a revised version of the manuscript that addresses the points raised during the review process.

1. Please be consistent with abbreviations- sometimes T2D is used whereas type2 diabetes is spelled out elsewhere.

2. Carefully proof-read for typos

3. Regarding classification of income groups into "poor/non-poor", i think ascribing poverty without justification may be inapropriate. Consider using a more justifiable terminology for the classification.

We look forward to receiving your revised manuscript.

Kind regards,

Ejemai Eboreime, MD, MSc, PhD

Academic Editor
---

## [Editor Report · Decision Letter 2]

21 Jul 2023

Catastrophic health care spending in managing type 2 diabetes before and during the COVID-19 pandemic in Tanzania

PGPH-D-23-00508R2

Dear Dr Binyaruka,

We are pleased to inform you that your manuscript 'Catastrophic health care spending in managing type 2 diabetes before and during the COVID-19 pandemic in Tanzania' has been provisionally accepted for publication in PLOS Global Public Health.

Best regards,

Ejemai Eboreime, MD, MSc, PhD

Academic Editor